# Sitagliptin Mitigates Diabetic Nephropathy in a Rat Model of Streptozotocin-Induced Type 2 Diabetes: Possible Role of PTP1B/JAK-STAT Pathway

**DOI:** 10.3390/ijms24076532

**Published:** 2023-03-31

**Authors:** Sarah M. AL-Qabbaa, Samaher I. Qaboli, Tahani K. Alshammari, Maha A. Alamin, Haya M. Alrajeh, Lama A. Almuthnabi, Rana R. Alotaibi, Asma S. Alonazi, Anfal F. Bin Dayel, Nawal M. Alrasheed, Nouf M. Alrasheed

**Affiliations:** 1PharmD. Program, College of Pharmacy, King Saud University, Riyadh 11451, Saudi Arabia; 2Department of Pharmacology and Toxicology, College of Pharmacy, King Saud University, Riyadh 11451, Saudi Arabia

**Keywords:** diabetic nephropathy, Janus kinase/signal transducer activator of transcription pathway, protein tyrosine phosphatase 1B, inflammation

## Abstract

Diabetic nephropathy (DN) is a microvascular complication of diabetes mellitus. This study examined the therapeutic effects of sitagliptin, a dipeptidyl peptidase inhibitor, on DN and explored the underlying mechanism. Male Wistar albino rats (*n* = 12) were intraperitoneally administered a single dose of streptozotocin (30 mg/kg) to induce diabetes. Streptozotocin-treated and untreated rats (*n* = 12) were further divided into normal control, normal sitagliptin-treated control, diabetic control, and sitagliptin-treated diabetic groups (*n* = 6 in each). The normal and diabetic control groups received normal saline, whereas the sitagliptin-treated control and diabetic groups received sitagliptin (100 mg/kg, p.o.). We assessed the serum levels of DN and inflammatory biomarkers. Protein tyrosine phosphatase 1 B (PTP1B), phosphorylated Janus kinase 2 (P-JAK2), and phosphorylated signal transducer activator of transcription (P-STAT3) levels in kidney tissues were assessed using Western blotting, and kidney sections were examined histologically. Sitagliptin reduced DN and inflammatory biomarkers and the expression of PTP1B, p-JAK2, and p-STAT3 (*p* < 0.001) and improved streptozotocin-induced histological changes in the kidney. These results demonstrate that sitagliptin ameliorates inflammation by inhibiting DPP-4 and consequently modulating the PTP1B-related JAK/STAT axis, leading to the alleviation of DN.

## 1. Introduction

Diabetic nephropathy (DN) is a microvascular complication of diabetes mellitus and is a major contributor to morbidity and mortality worldwide [1]. Although the pathophysiology of diabetes-induced nephropathy is not fully understood, studies have reported a strong association between chronic hyperglycemia-induced oxidative stress and the hyperlipidemia-induced release of pro-inflammatory cytokines with diabetes mellitus development and progression [2,3]. A previous study showed that rats with diabetes produced significantly higher levels of interleukin (IL-1) and tumor necrosis factor-alpha (TNF-α) than those without diabetes, suggesting the involvement of pro-inflammatory cytokines in the development of DN [4]. Furthermore, the dysregulation of the synthesis of incretin, which stimulates insulin secretion in response to glucose intake, has been shown to be associated with DN development [5]. The efficacy of incretin-based therapies against DN has been reported [6]. Dipeptidyl peptidase-4 (DPP4) deactivates incretin hormones, whereas DPP4 inhibitors, such as sitagliptin, block this degradation, increase the incretin levels, and prolong insulin secretion [7]; therefore, sitagliptin could be a potential therapeutic for type 2 diabetes mellitus. Furthermore, recent studies have shown that DPP-4 inhibitors also have anti-inflammatory effects that provide endothelial protection [8,9,10,11,12,13].

Protein tyrosine phosphatase (PTP) 1B (PTP1B) is one of the most common PTPs and is a critical positive and negative regulator of various signaling pathways. PTP1B plays a fundamental role in controlling glucose uptake and insulin regulation, making it a promising therapeutic target for type 2 diabetes [14]. A study conducted in mice showed that podocyte PTP1B deficiency attenuates hyperglycemia-induced renal injury [15,16]. Furthermore, PTP1B disruption enhances renal insulin signaling and suppresses inflammation and fibrosis [17]. In addition, some studies have shown that PTP1B regulates the pro-inflammatory and anti-inflammatory effects of the Janus kinase-signal transducer and activator of transcription (JAK-STAT) signaling pathways in hematological diseases [16]. The JAK-STAT pathway is a signaling mechanism for many cytokines and growth factors. JAK is activated by multiple immune cytokines, which in turn stimulate cellular events, including proliferation, differentiation, migration, and apoptosis [18,19]. The activation of the JAK-STAT pathway has been shown to stimulate the excessive growth of glomerular mesangial cells, which contributes to DN development [20]. The JAK-STAT pathway plays an important role in inflammatory diseases, and JAK inhibitors have been shown to be effective in alleviating inflammation. Crucial for controlling inflammation in diabetic kidney diseases, STAT3 has been reported to be activated early in DN [21]. These studies suggest a link between PTP1B, a negative regulator of the JAK-STAT signaling pathway [14], inflammation, and the development of DN [22]. 

Furthermore, a few studies suggest that sitagliptin reduces inflammation, which plays an important role in the pathogenesis of DN. We hypothesized an association between DPP-4 inhibition and the PTP1B/JAK-STAT pathway in the alleviation of DN. To test this hypothesis, in this study, we aimed to evaluate the therapeutic potential of sitagliptin in attenuating DN and explore the underlying mechanism.

## 2. Results

### 2.1. Sitagliptin Reverses Hyperglycemia and Weight Loss and Alleviates DN in Diabetic Rats

#### 2.1.1. Effects on Blood Glucose Level

After inducing diabetes using streptozotocin (STZ; 30 mg/kg; intraperitoneally), rats with or without diabetes were treated with or without sitagliptin (100 mg/kg; oral gavage), as shown in Figure 1. The results demonstrated significantly elevated glucose levels in the STZ control group (untreated, with diabetes) compared to those in the normal control (untreated, without diabetes) and normal sitagliptin groups (treated, without diabetes) (*p* < 0.001; Table 1). The rats in the STZ sitagliptin group (treated, with diabetes) showed a significant reduction in glucose levels compared to those in the STZ control group (*p* < 0.001) (Appendix A).

#### 2.1.2. Effects on Kidney-Weight-to-Body-Weight Ratio 

The kidney-weight-to-body-weight ratio is considered an indicator of the development of diabetic renal hypertrophy [23]. This ratio was significantly higher in the STZ control group than that in the normal control group (*p* < 0.001). On the contrary, sitagliptin treatment alleviated the STZ-induced increase in the kidney-weight-to-body-weight ratio in diabetic rats (*p* < 0.001) (Table 1) (Appendix A).

#### 2.1.3. Effects on Serum Creatinine, Blood Urea Nitrogen (BUN), and Urea Levels

Serum creatinine, blood urea nitrogen (BUN), and urea levels were measured to monitor renal excretory function. As shown in Table 1, a significant increase in serum creatinine was observed in the STZ control group compared to that in the normal control group (*p* < 0.01), which indicated that renal function was markedly impaired in the former. However, this increase was alleviated by sitagliptin treatment in diabetic rats (*p* < 0.05). In addition, the STZ sitagliptin group showed a decrease in urea and BUN levels compared to the STZ control group (*p* < 0.001), suggesting that treatment with sitagliptin attenuated DN. The BUN and urea levels in the STZ control group were significantly higher than those in the normal control and normal sitagliptin groups (*p* < 0.01 in both), indicating the successful induction of DN (Appendix A).

### 2.2. Sitagliptin Alleviates STZ-Induced Alterations in Inflammatory Biomarkers in Experimental Diabetic Rats

#### 2.2.1. Effects on IL-6

As shown in Figure 2, the expression of IL-6 in the STZ control group (*p* < 0.05) was significantly higher than that in the normal control and normal sitagliptin groups. The significant elevation of serum IL-6 levels in the STZ control group was alleviated with sitagliptin treatment in the STZ sitagliptin group (*p* < 0.001) (Appendix A). 

#### 2.2.2. Effects on TNF-α Levels

As shown in Figure 3, the plasma level of TNF-α was markedly higher in the STZ control group than that in the normal control (*p* < 0.05). However, TNF-α levels significantly decreased after sitagliptin treatment in the STZ sitagliptin group compared to those in the STZ control group (*p* < 0.001) (Appendix A).

### 2.3. Effects of Sitagliptin on Protein Expression Levels in Experimental Diabetic Rats Using Western Blot Analysis

To investigate the effect of sitagliptin on phosphorylated JAK2 (P-JAK2), phosphorylated STAT3 (P-STAT3), and PTP1B in the kidney, the concentrations and distributions of these proteins were evaluated using Western blotting.

#### 2.3.1. Effects on P-JAK2 Expression

As illustrated in Figure 4, the expression of P-JAK2 was significantly higher in the diabetic groups than that in the non-diabetic groups (*p* < 0.001), whereas that in the STZ control group was the highest. Sitagliptin administration significantly decreased the expression of P-JAK2 in the STZ sitagliptin group compared to that in the STZ control group (*p* < 0.001) (Appendix A).

#### 2.3.2. Effects on P-STAT3 Expression

As illustrated in Figure 5, the expression of P-STAT3 in the diabetic control group was significantly higher than that in the non-diabetic control and sitagliptin-only non-diabetic groups (*p* < 0.001 in both). The expression of P-STAT3 in the diabetic group treated with sitagliptin was markedly downregulated compared to that in the diabetic control group (*p* < 0.001) (Appendix A).

#### 2.3.3. Effects on PTP1B Expression

Immunoblot data showed a significant increase in PTP1B expression in STZ-induced diabetic rats compared with non-diabetic rats (*p* < 0.001). As shown in Figure 6, PTP1B expression in diabetic rats treated with sitagliptin was significantly reduced compared to that in the diabetic control group (*p* < 0.001) (Appendix A).

### 2.4. Histopathological Changes in Experimental Diabetic Rat Kidneys Treated with Sitagliptin

A histological examination of rat kidney sections stained with hematoxylin and eosin (H&E) was performed. The non-diabetic control rats had normal kidney structures with regular glomeruli and renal tubules (Figure 7A), whereas STZ induced the thickening of the glomerular capsule space, necrosis, diffuse interstitial inflammation and hemorrhage, and severe glomerular lesions (Figure 7B). Like the non-diabetic control group, the diabetic sitagliptin group also showed normal kidney structures with regular glomeruli and renal tubules (Figure 7C). Furthermore, sitagliptin improved the kidney structure by ameliorating almost all lesions in diabetic rats treated with sitagliptin (Figure 7D).

## 3. Discussion

The prevalence of diabetes has increased in recent decades [24]. Approximately 40% of all patients with diabetes experience kidney failure, making diabetes a primary factor in end-stage renal disease [25]. The only currently approved treatment for DN is the blockade of the renin–angiotensin system (RAS), which appeared to be effective 20 years ago. However, this treatment has various limitations and does not result in significant benefits for patients with DN [26,27]. Therefore, identifying potential new therapies and molecular targets is of immense practical importance. 

A single dose of STZ is commonly used to establish rodent models of DN within 4–8 weeks [23,28]. STZ is a synthetic antineoplastic agent that is classified as an anti-tumor antibiotic. The diabetogenic effect of STZ is driven by the production of reactive oxygen species, which results in the necrosis of pancreatic β-cells, leading to hyperglycemia and culminating in diabetic renal damage. In addition, STZ exerts direct renal toxicity. This makes it difficult to distinguish whether renal dysfunction occurs due to its direct toxic effects or is induced by DN. Therefore, studies examining the effects of treatments on the occurrence of DN should not commence until at least 3 weeks after STZ administration, when the kidneys have recovered from the acute mild nephrotoxic effects of STZ [29,30,31]. In this study, STZ and a high-fat diet were used to induce diabetes accompanied by DN. STZ-treated rats exhibited hyperglycemia, as indicated by increased fasting blood sugar levels. Consistent with a previous study [23], STZ increased the kidney-weight-to-body-weight ratio and the levels of serum creatinine, urea, BUN, and other DN biomarkers. Moreover, histopathological changes in the kidneys of STZ-induced diabetic rats, including the thickening of the glomerular membrane, necrosis, inflammation, and glomerular lesions, confirmed DN. Concordant with those reported in previous studies [32,33,34,35], these data provide evidence for the successful establishment of the DN model by STZ treatment.

In this study, we examined the protective effects of sitagliptin, a DPP-4 inhibitor, against the development of DN because DPP-4 is highly expressed in epithelial cells, such as renal glomeruli [36]. In addition, DPP-4 is highly expressed in individuals with DN [37,38]. The level of DPP-4 is increased in patients with type 2 diabetes and corresponds to the onset of albuminuria [39]. An earlier study reported that circulating levels of creatinine and BUN were reduced in DPP-4 mutant rats, suggesting that DPP-4 inhibitors protect against kidney injury, especially in individuals with type 2 diabetes mellitus [40]. Moreover, it has also been shown that sitagliptin exerts its activity by inhibiting the DPP-4 enzyme, which increases the active GLP-1 concentration, resulting in the regulation of glucose homeostasis. Although the mechanism by which sitagliptin attenuates DN remains unclear, some studies have demonstrated that DPP-4 inhibitors exert renoprotective effects by decreasing kidney inflammation and podocyte injury [39]. In addition, sitagliptin reduces albuminuria by controlling hyperglycemia [37,41,42]. Furthermore, augmented levels of GLP-1, mediated by the inhibition of DPP-4, exert anti-inflammatory actions in hyperglycemia, which may mitigate DN [43]. Therefore, by increasing the concentration of active GLP-1, sitagliptin may synergistically reduce inflammation in diabetic kidneys [44,45]. It has also been shown that GLP-1 receptors are downregulated under diabetic conditions, whereas DPP-4 is upregulated in the renal glomeruli in diabetic glomerulosclerosis [8,38,46]. In this study, sitagliptin decreased hyperglycemia and ameliorated DN by reducing the serum levels of urea, BUN, and creatinine. In addition, after three weeks of sitagliptin administration in the STZ model of type 2 diabetes, rat kidney lesions and structures were improved, with a concomitant reduction in inflammatory markers. These results are consistent with those of previous studies [12,23,38,47].

A previous study reported that sitagliptin exerted anti-inflammatory and anti-apoptotic effects [45]. However, whether the anti-inflammatory effects protect against inflammation and the proapoptotic state in the kidneys of diabetic rats is not clear. In the present study, we investigated the therapeutic potential of sitagliptin in DN and the possible related signaling pathways, particularly inflammation, to understand this. In a randomized, placebo-controlled, double-blinded prospective study of obese patients with type 2 diabetes, sitagliptin exerted a comprehensive and potent anti-inflammatory effect, as demonstrated by a rapid reduction in the expression of TNF-α, IL-6, and other inflammatory cytokines, which may have contributed to the inhibition of atherosclerosis [48]. The findings of this study demonstrated that sitagliptin alleviated inflammation in the kidneys and markedly decreased the levels of IL-6 and TNF-α in diabetic rats. These findings indicate that sitagliptin effectively ameliorated inflammation and improved kidney function.

The JAK/STAT signaling pathway comprises three components: tyrosine kinase-associated receptors, tyrosine Janus kinase, and signal transducer and activator of transcription. Over the last three decades, several studies have shown the involvement of the JAK/STAT signaling pathway in the development of different diseases, especially inflammatory diseases and cancers [49,50]. Previous studies have also emphasized the crucial role of the negative regulation of JAK/STAT in inhibiting hyperglycemia-induced renal damage and improving renal function by reducing renal inflammation and fibrotic lesions and slowing DN progression [51,52]. A previous study in a type 2 diabetic rat model treated with sitagliptin revealed the cardioprotective effect of sitagliptin [53]. In this study, sitagliptin modulated the JAK/STAT pathway by reducing IL-6, PJAK-2, and PSTAT3 levels. These findings, in line with those of previous studies, indicate that sitagliptin attenuates DN via the JAK/STAT signaling pathway.

Zabolotny et al. [54] reported that inflammation regulates the overexpression of PTP1B and that the activation of the pro-inflammatory transcription factor nuclear factor-kB by TNF-α increases *PTP1B* mRNA and protein expression. Another study reported that the myeloid-specific deletion of PTP1B in mice protects against both acute and chronic inflammation induced by lipopolysaccharides or a high-fat diet [16]. The present study explored the link between PTP1B and inflammation and shed light on the role of the PTP1B/JAK2/STAT3 axis in triggering DN.

Protein tyrosine kinases and PTPs are the main regulators of protein tyrosine phosphorylation and dephosphorylation [54]. PTP1B is a negative regulator of insulin receptors and is ubiquitously expressed in various metabolic tissues, particularly the liver, skeletal muscle, and adipose tissues [55]. In recent years, PTP1B has attracted considerable attention as a potential therapeutic target in metabolic syndromes [56,57,58]. A previous study on mice with podocyte PTP1B disruption demonstrated that PTP1B deficiency improved hyperglycemia-induced albuminuria and renal injury and reduced inflammation and fibrosis [15]. Another study in PTP1B knockout mice revealed that a deficiency in PTP1B alleviated hypothalamic inflammation via the activation of the JAK2/STAT3 signaling pathway in microglial cells [59]. In this study, we demonstrated that sitagliptin upregulated PTP1B, P-JAK2, and PSTAT3, indicating that sitagliptin exhibits an anti-inflammatory effect by inhibiting the PTP1B/JAK-STAT signaling pathway in diabetic rats. The latter is associated with inflammation, which ultimately leads to the development of DN. 

This study provides insights into the renoprotective effects of sitagliptin in an STZ rat model of type 2 diabetes. Based on the findings of this study, we propose that sitagliptin ameliorates inflammation by inhibiting DPP-4 and consequently modulating the PTP1B-related JAK/STAT axis, leading to the alleviation of DN.

Study limitations:

Although each step in the Western blot protocol was optimized to ensure reliable and reproducible results, the immunoblot images of PTP1B and STAT3 were not well defined, which might have affected the reliability of our findings. These unclear images may be attributed to using the total tissue lysate in an in vivo animal model. In addition to natural variations from animal to animal, an issue confounding in vivo studies is that target protein expression in signaling pathways is activated for a while but is then deactivated or degraded. This limitation could be avoided by using an adjunct in vitro group, as has been implemented in animal studies measuring cell lysates in vitro and tissue lysates in vivo [56,58,60]. Employing an in vitro model in future investigations may improve the results’ reliability. 

## 4. Materials and Methods

### 4.1. Drugs, Chemicals, and Antibodies

STZ was purchased from Sigma-Aldrich (St. Louis, MO, USA). Sitagliptin (Januvia) was purchased from Merck & Co., Inc. (Whitehouse, NJ, USA). The inflammatory biomarkers TNFα and IL-6 and DN biomarkers urea, BUN, and creatinine were measured using specific enzyme-linked immunosorbent assay (ELISA) kits from Abcam (Biotechnology Inc., Cambridge, UK). The rabbit polyclonal PTP1B antibody and anti-JAK2, anti-P-JAK2, anti-STAT3, and anti-P-STAT3 antibodies were obtained from Santa Cruz Biotechnology, Inc. (Dallas, TX, USA). Both β-actin, the housekeeping loading control antibody, and other associated primary antibodies were bought from Abcam (Biotechnology Inc., Cambridge, UK). Anti-rabbit, anti-mouse, anti-goat horseradish peroxidase (HRP)-conjugated secondary antibodies, glucose assay kit were sourced from Sigma-Aldrich (St. Louis, MO, USA). Different analytical-grade chemicals and reagents were sourced from standard commercial firms. 

### 4.2. Experimental Animals

Male Wistar albino rats (8–10 weeks old) weighing 200–250 g were obtained from the Animal Care Center, College of Pharmacy, King Saud University, Saudi Arabia. The rats were properly housed, inspected daily, and kept in special cages at monitored constant temperature (22–23 °C), humidity (60%), and light–dark cycle (12/12 h). During the trial, the animals were provided distilled water ad libitum and rodent maintenance chow. The rats in the control group were fed a normal diet, whereas the rats in the experimental groups were fed a high-fat diet (containing a mixed meal with 63% calories, 1% (*w*/*w*) sucrose, 1% (*w*/*w*) cholesterol, high levels of protein, and 25% extra virgin olive oil), with some alterations to induce obesity and insulin resistance. The study protocol complied with the recommendations of the Experimental Animals Ethics Committee Acts of King Saud University, Research Ethics Committee (REC) (Ethics Reference No. KSU-SE-20-15). The dose regimen used in this study was based on preliminary studies conducted in other laboratories.

### 4.3. Induction of Diabetes

The rats were subjected to 24 h fasting. Afterward, a single dose of STZ (30 mg/kg) [12,61] diluted in 0.1 M citrate buffer (pH 4.5) immediately before injection was administered intraperitoneally to induce type 2 diabetes in the rats. Seventy-two hours after the STZ injection, a drop of blood was collected from the tail vein, and random blood glucose levels were measured using the Accu-Chek Advantage II blood glucose monitoring system (Roche Diagnostic, Indianapolis, IN, USA). Diabetes was confirmed when the blood glucose levels were >20 mmol/L (200 mg/dL).

### 4.4. Experimental Design and Measurements

Initially, the weights of 24 rats were determined, and they were assigned to one of two groups (*n* = 12 each): one was treated with STZ as described above, and the other was used as the control. The rats in these two groups were further divided into four groups (*n* = 6 in each group) depending on their treatment regime: the non-diabetic control received normal saline (0.9% NaCl; drug vehicle) for six weeks; diabetic untreated rats received normal saline (0.9% NaCl) for six weeks; the non-diabetic treatment control received normal saline for two weeks, followed by sitagliptin treatment (100 mg/kg/day) [62,63] for four weeks; and diabetic rats that received sitagliptin treatment were given normal saline for two weeks, followed by four-week sitagliptin treatment (100 mg/kg/day). The administration of all treatments was performed via oral gavage every day. Figure 1 illustrates the experimental approach adopted. 

The body weights of the rats were recorded weekly until the end of the experiment. Rats were placed in metabolic cages at weeks 0, 3, and 6, and the amount of urine voided during a 24 h period was measured. After measuring its volume, the collected urine was deep frozen (−80 °C). At the end of the experiment, all rats were fasted overnight (12 h) and euthanized by gradually increasing the concentration of carbon dioxide (CO_2_) and decapitated. The collection and processing of blood samples were performed for the purpose of separating serum so it could be biochemically analyzed. The immediate removal of the kidneys was followed by rinsing them in ice-cold phosphate-buffered saline and then weighing them. This enabled the calculation of the ratio of kidney weight to body weight. Subsequently, the dissection of some of the kidney tissues was performed, followed by their collection in neutral buffered formalin (4%) for histological and immunohistochemical processing, while the remaining tissues were kept at −80 °C for molecular analysis [64].

### 4.5. Measurement of Serum Glucose

Serum glucose levels were determined using a glucose analytical assay kit [65] according to the manufacturer’s instructions.

### 4.6. Determination of DN Biomarkers

Kidney biomarkers (blood urea nitrogen (BUN), creatinine, and urea) in serum samples were determined using specific rat ELISA kits according to the manufacturer’s specifications. With regard to albumin, creatinine, and urea assays, the incubation of 50 µL of all standards or samples was performed for 2 h at 37 °C. Washing of the wells was conducted to ensure that all unbound substances were removed, followed by the addition of 50 µL of the chosen primary detector antibody to every well and then further incubation for 1 h at 37 °C. Afterward, 50 µL of the HRP-conjugated secondary antibody was added to each well and incubated for 1 h at 37 °C. Subsequently, 100 µL of conjugated development solution was introduced to each of the wells and incubated in the dark for 30 min at 37 °C. Finally, 100 µL of stop solution was added to each well, and the absorbance was recorded at 450 nm using a microplate reader (BioTek Instruments, Winooski, VT, USA). The concentrations in the samples were calculated using a standard curve following the assay kit instructions.

### 4.7. Assessment of Inflammatory Biomarkers

Serum was separated from aliquots of the blood sample using ELISA kits, following the manufacturer’s instructions (Abcam, Biotechnology Inc., Cambridge, UK), to measure serum levels of IL-6 and TNF-α.

### 4.8. Western Blot Analysis

The homogenization of frozen samples of kidney tissues was performed in ice-cold lysis buffer prior to transfer to an immunoprecipitation assay (RIPA) buffer, which contained protease and phosphatase inhibitor cocktails at equal volumes. A direct detection qualitative assay method was used to determine the concentrations of proteins. In summary, sulfate–polyacrylamide gel electrophoresis was used to separate 60 µg of the homogenized protein sample and was followed by the transfer of proteins to polyvinylidene difluoride membranes (0.2 µm, Immun-Blot^®^, Bio-Rad, Hercules, CA, USA). The membranes were then blocked using a 5% mixture of non-fat dry milk and bovine serum albumin solution and incubated for 1 h at room temperature. Subsequently, the membranes were incubated overnight at 4 °C with primary antibodies for PTP1B, P-JAK2, and P-STAT3 diluted in Tris-buffered saline and Tween (TBST) buffer (1:1000); β-actin (1:2000) was used as a housekeeping loading control antibody. The membranes were washed and incubated for 1 h at room temperature with an HRP-conjugated anti-rabbit (1:5000) secondary antibody diluted in TBST buffer. The protein blots were developed using an enhanced chemiluminescence detection kit (GE Healthcare, Buckingham, Buckinghamshire, UK) for 2 min before image acquisition. Immunoreactive bands were visualized using Image Qurant LAS 400 (GE Healthcare, Buckingham, Buckinghamshire, UK). The intensities of various protein bands were quantified densitometrically using Image J software (NIHI Image, Bethesda, MD, USA) and normalized against the loading control (β-actin) by dividing the value of the target protein by that of β-actin. The relative values were normalized to the control, whose value was arbitrarily fixed to 1, and expressed as fold induction.

### 4.9. Histological Examination

The rats were then exposed to CO_2_ for anesthesia, and the kidneys were immediately removed, washed with ice-cold saline, and carefully cleaned of extraneous fat and connective tissue. Each kidney was placed directly in 10% neutral formalin for 24 h, after which they were dehydrated in high concentrations of ethanol, cleared with xylene, and embedded in paraffin. Paraffin-embedded sections (4 µm) were prepared for morphological examination using H&E staining. High-resolution digital cans of all stained slides were generated with a Leica Biosystems Aperio digital Scanner (Leica Biosystems (Aperio), Vista, CA, USA) and were viewed using Aperio’s viewing and image analysis tools. The images were analyzed by investigators specializing in this field.

### 4.10. Statistical Analysis

All data are presented in form of mean ± standard error of the mean (SEM). Groups were compared statistically using one-way analysis of variance (ANOVA) and then the Tukey–Kramer post hoc test. GraphPad Prism Windows software version 9 was used for all statistical analyses (GraphPad Software Inc., San Diego, CA, USA), and *p* < 0.05 was accepted as the level of statistical significance.

## 5. Conclusions

In conclusion, the findings of this study indicate that sitagliptin exerts renoprotective effects in STZ-induced diabetic rats. These renoprotective effects include the attenuation of inflammation via the modulation of PTP1B and the JAK/STAT signaling pathway, contributing to the mitigation of DN. The results suggest that sitagliptin can be considered a promising drug for DN treatment. In addition, this study provides initial evidence of the crucial role of the PTP1B and JAK/STAT signaling pathways in the inflammation-driven development of DN.

## Figures and Tables

**Figure 1 ijms-24-06532-f001:**
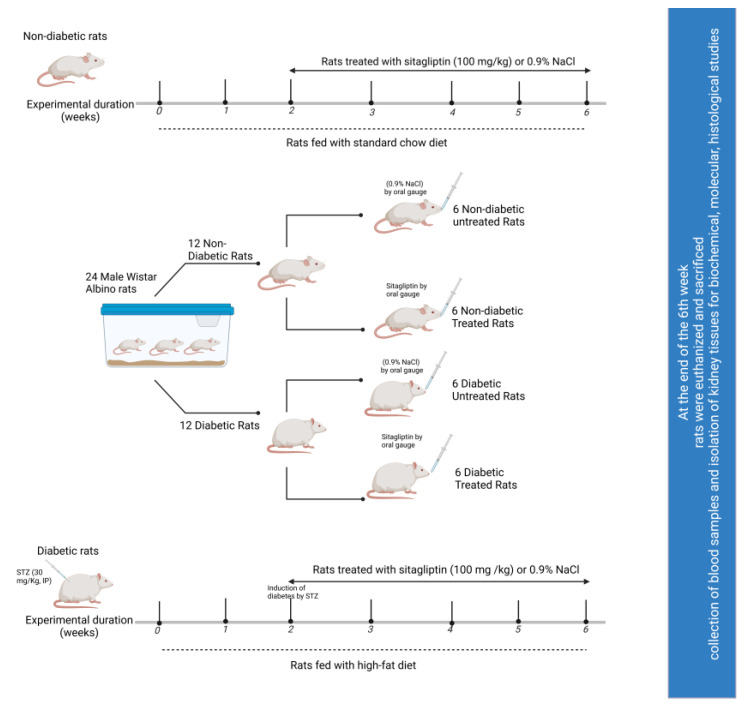
Schematic representation of the experimental design.

**Figure 2 ijms-24-06532-f002:**
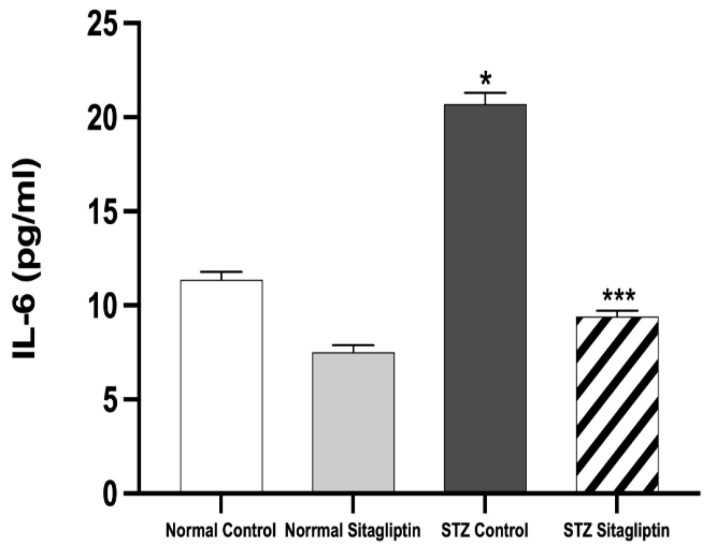
Effects of sitagliptin on interleukin-6 (IL-6) plasma levels in STZ-induced diabetic rats. Data are expressed as means ± SEM, according to a one-way ANOVA followed by the Bonferroni test (*n* = 6 animals per group). * *p* < 0.05 vs. normal control (non-diabetic control); *** *p* < 0.001 vs. STZ control group; STZ, streptozotocin.

**Figure 3 ijms-24-06532-f003:**
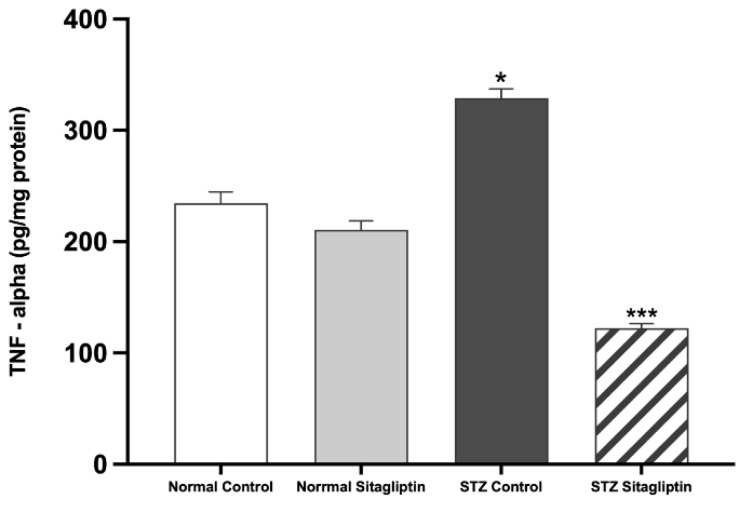
Effects of sitagliptin on plasma tumor necrosis factor-alpha (TNF-α) levels in STZ-induced diabetic rats. Data are expressed as means ± SEM, according to one-way ANOVA followed by the Bonferroni test (*n* = 6 animals per group). * *p* < 0.05 vs. normal control (non-diabetic control); *** *p* < 0.001 vs. STZ control group; STZ, streptozotocin.

**Figure 4 ijms-24-06532-f004:**
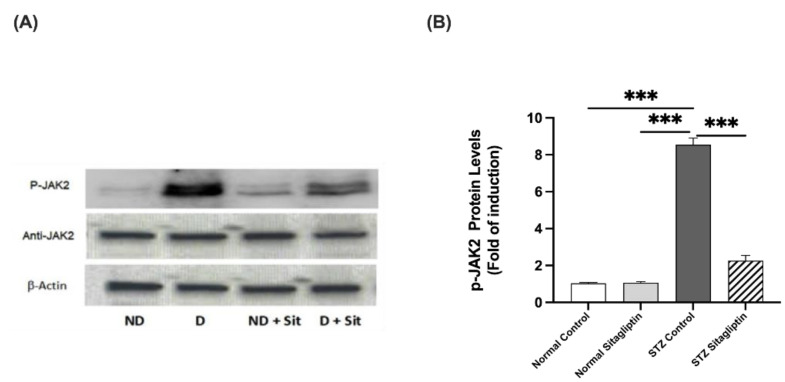
Effects of sitagliptin on phosphorylated Janus kinase (P-JAK2) expression in STZ-induced diabetic rats. (**A**) Representative images of P-JAK2 in kidney tissue homogenates; ND, normal control group; D, STZ-induced diabetic control group; ND + Sit, normal control group treated with sitagliptin; D + Sit, sitagliptin-treated diabetic group. Resolution of the proteins obtained from sample tissues was performed with SDS-PAGE, and then the P-JAK antibody was used for immunoblotting, as well as total JAK2 and β-actin antibodies subsequent to stripping. (**B**) Quantitative outcomes of immunoblotting technique utilizing ImageJ software. Β-Actin was used for the normalization of the levels of P-JAK2 protein. The expression of quantities is in the form of fold induction relative to the group comprising normal controls. The data denote the mean ± SEM (*n* = 6 samples from 6 rats per group). The one-way ANOVA test and then the Bonferroni post hoc test were used to compare groups; *** *p* < 0.001.

**Figure 5 ijms-24-06532-f005:**
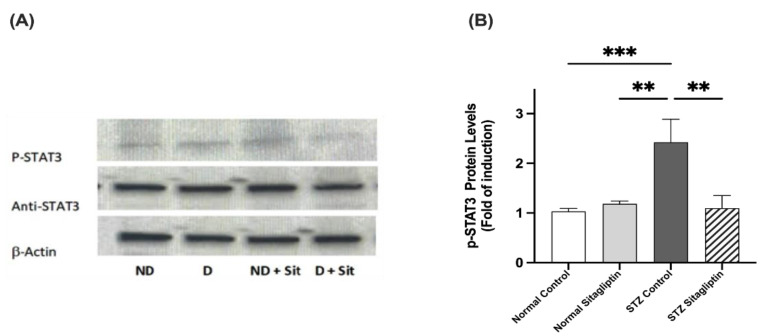
Effects of sitagliptin on phosphorylated signal transducer and activator of transcription (P-STAT3) expression in STZ-induced diabetic rats. (**A**) Representative images of P-JAK2 in kidney tissue homogenates; ND, normal control group; D, STZ-induced diabetic control group; ND + Sit, normal control group treated with sitagliptin; D + Sit, sitagliptin-treated diabetic group. Resolution of the proteins obtained from sample tissues was performed by using SDS-PAGE, and then the P-STAT3 antibody was used for immunoblotting, along with total STAT3 and β-actin antibodies subsequent to stripping. (**B**) ImageJ software was used to quantify the immunoblot densities. Β-Actin was used to normalize the levels of P-STAT3. The expression of quantities is indicated as fold induction relative to the group comprising normal controls. Data represent the mean ± SEM (*n* = 6 samples from 6 animals per group). The one-way ANOVA test and then the Bonferroni post hoc test were used to compare groups; ** *p* < 0.01, *** *p* < 0.001.

**Figure 6 ijms-24-06532-f006:**
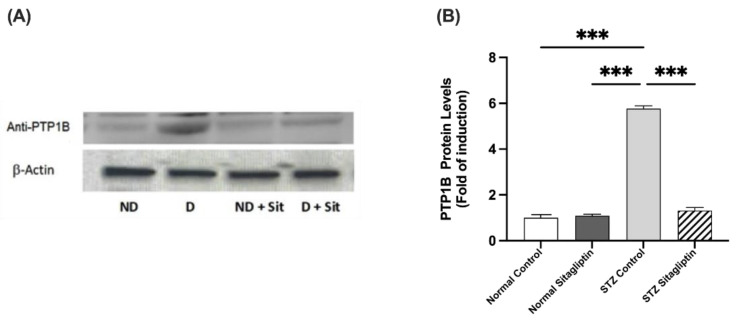
Effects of sitagliptin on protein tyrosine phosphatase 1B (PTP1B) expression in STZ-induced diabetic rats. (**A**) Images representing the immunoblotting of levels of proteins in the homogenates of rat kidney tissues; ND, normal control group; D, STZ-induced diabetic control group; ND + Sit, normal control group treated with sitagliptin; D + Sit, sitagliptin-treated diabetic group. Resolution of the proteins obtained from the sample tissues was performed using SDS-PAGE, and then the PTP1B antibody was used for immunoblotting, along with the β-actin antibody subsequent to stripping. (**B**) ImageJ software was used to quantify the immunoblot band densities. β-Actin was utilized to normalize the levels of PTP1B protein. The expression of quantities is indicated as fold induction relative to the group comprising normal controls. Data denote the mean ± SEM (*n* = 6 samples from six animals per group). The one-way ANOVA test and then the Bonferroni post hoc test were used to compare groups; *** *p* < 0.001.

**Figure 7 ijms-24-06532-f007:**
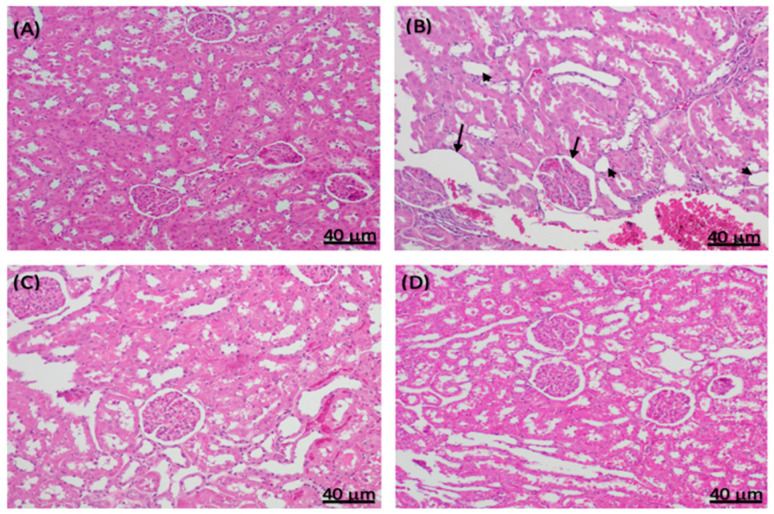
Effect of sitagliptin on STZ-induced pathological changes in rat kidney sections stained with hematoxylin and eosin. Representative images of kidney sections in (**A**) normal control rats with a normal histological appearance of renal corpuscles and tubules; (**B**) diabetic STZ control rats with increased thickening of the basement membrane (black arrows), necrosis (arrowheads), and diffuse interstitial inflammation and hemorrhage; (**C**) non-diabetic normal rats treated with sitagliptin (100 mg/kg/day, p.o.) with a normal basement membrane of the renal corpuscles; and (**D**) diabetic STZ rats treated with sitagliptin (100 mg/kg/day, p.o.), showing the alleviation of the STZ-induced damage.

**Table 1 ijms-24-06532-t001:** Effects of sitagliptin on DN biomarkers in STZ-induced diabetic rats.

	Glucose (mg/dL)	Kidney Weight (g)	Kidney/Body Weight Ratio (%) (mg/g)	Urea (mg/dL)	BUN (mg/dL)	Creatinine (IU/L)
Normal control	3.72 ± 0.58	1.83 ± 0.16	0.57 ± 0.05	11.92 ± 2.85 **	5.56 ± 1.33 ^$$^	0.28 ± 0.03
Normal sitagliptin	3.98 ± 0.25	2.00 ± 0.06	0.74 ± 0.03	9.65 ± 1.13 **	4.50 ± 0.53 ^$$^	0.25 ± 0.44
STZ control	9.21 ± 1.41 ***	2.80 ± 0.25 **	1.04 ± 0.08 ***	49.70 ± 4.02 ***	23.19 ± 1.87 ***	0.64 ± 0.10 **
STZ sitagliptin	4.26 ± 1.06 ^###^	1.87 ± 0.05 ^##^	0.68 ± 0.02 ^###^	35.31 ± 10.67 ^###^	16.47 ± 4.98 ^###^	0.31 ± 0.08 ^#^

Data are expressed as mean ± standard error of the mean (SEM) (*n* = 6 animals per group). ** *p* < 0.01, *** *p* < 0.001 vs. normal control (non-diabetic control); ^#^
*p* < 0.05, ^##^
*p* < 0.01, ^$$^
*p* < 0.01, ^###^
*p* < 0.001 vs. STZ control group. DN, diabetic nephropathy; STZ, streptozotocin.

## Data Availability

The data presented in this study are available on request from the corresponding author.

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
