# Peer review of "Sitagliptin Mitigates Diabetic Nephropathy in a Rat Model of Streptozotocin-Induced Type 2 Diabetes: Possible Role of PTP1B/JAK-STAT Pathway"

_ijms, 2023, doi:10.3390/ijms24076532_

Round 1

Reviewer 1 Report

In this manuscript the authors contribute to shed light on a potential use of Sitagliptin ( a dipeptidyl peptidase inhibitor)  therapy in attenuating the diabetic neuropathy (DN), a common microvascular complication in diabetes characterized by highs levels of several inflammatory molecules. Therefore, the authors conducted an experimental animal study to test the possible therapeutic effects of Sitagliptin and, also, they want to explore the underlying mechanism. I think that the manuscript should be accepted for publication, I have only few minor clarifications/changes to request authors that could improve the manuscript.

Q1: I think that the authors forgot to include some references in the text. In detail, the references could be added in the following lines: 63(pg. 2), 244 (pg.8) and 362 (pg. 10). Furthermore, I think that a reference between the link of the role of the JAK/STAT signaling pathway in the development of inflammatory disease, such as cancer, could be insert at the line 269 (pg. 8).

Q2:  The authors assert that the protein bands are quantified densitometrically using a software. The images of western blot results (Figure S1) are not very well defined, particularly for PTP1B and STAT3. Could this have affected the reliability of the results? If yes I think that this limit should be mentioned in the discussion.

Q3:  The protein STAT3 is mentioned directly in the discussion section of the text, I advise the authors to include a short sentence of its role in the introduction. This could help the readers better understand the molecular pathway described.

Author Response

Reviewer#1

In this manuscript the authors contribute to shed light on a potential use of Sitagliptin ( a dipeptidyl peptidase inhibitor)  therapy in attenuating the diabetic neuropathy (DN), a common microvascular complication in diabetes characterized by highs levels of several inflammatory molecules. Therefore, the authors conducted an experimental animal study to test the possible therapeutic effects of Sitagliptin and, also, they want to explore the underlying mechanism. I think that the manuscript should be accepted for publication, I have only few minor clarifications/changes to request authors that could improve the manuscript.

Response: Thank you very much.

Q1: I think that the authors forgot to include some references in the text. In detail, the references could be added in the following lines: 63(pg. 2), 244 (pg.8) and 362 (pg. 10). Furthermore, I think that a reference between the link of the role of the JAK/STAT signaling pathway in the development of inflammatory disease, such as cancer, could be insert at the line 269 (pg. 8).

Response: Thank you for your notification and suggestion. We have cited the references in the lines indicated in the manuscript.

Q2:  The authors assert that the protein bands are quantified densitometrically using a software. The images of western blot results (Figure S1) are not very well defined, particularly for PTP1B and STAT3. Could this have affected the reliability of the results? If yes I think that this  limit should be mentioned in the discussion.

Response: Thank you for your comment. We agree that the immunoblots for PTP1B and STAT3 are not well defined in Figure S1. We tried to optimize each step of protocol, including sample separation and gel electrophoresis, and we tried antibodies from different sources to ensure the results are reliable and reproducible, but, unfortunately, we had the same undefined (not clear) images. This might be attributed to our use of total tissue lysate from an in vivo animal model. In addition to animal variations, the protein expression for target proteins in signaling pathways is activated for a specific duration, and the protein then undergoes deactivation or degradation. This limitation could be avoided by investigating target proteins in vitro. Target proteins were investigated in several studies using cell lysates (in vitro) and tissue lysates (in vivo) from animal models [1-3]. Therefore, the addition of in vitro assays could improve the reliability of the results in our future study. We have addressed this in the study limitations (lines 305–315).

  1. Wang, Q., et al., STAT3 suppresses cardiomyocytes apoptosis in CVB3-induced myocarditis via survivin. Front Pharmacol, 2020. 11: p. 613883.
  2. Teimouri, M., et al., The role of protein tyrosine phosphatase 1B (PTP1B) in the pathogenesis of type 2 diabetes mellitus and its complications. J Physiol Biochem, 2022. 78(2): p. 307-322.
  3. Chen, Q., et al., TRIM18-Regulated STAT3 Signaling Pathway via PTP1B Promotes Renal Epithelial-Mesenchymal Transition, Inflammation, and Fibrosis in Diabetic Kidney Disease. Front Physiol, 2021. 12: p. 709506.

Q3:  The protein STAT3 is mentioned directly in the discussion section of the text, I advise the authors to include a short sentence of its role in the introduction. This could help the readers better understand the molecular pathway described?

Response: We appreciate your comment. We have addressed this in the Introduction section (lines 63–65).

"Activation of STAT3 has been reported to occur at an early stage of diabetic nephropathy. STAT3 is crucial for controlling the inflammation associated with diabetic kidney diseases."

Zheng et al. Inhibition of STAT3 in tubular epithelial cells prevents kidney fibrosis and nephropathy in STZ-induced diabetic mice. Cell Death Dis 2019. 10: p. 848.

Reviewer 2 Report

The topic is relevant, but I have a number of comments:

1)     Representatives of the class of DPP-4 inhibitors (glyptins) are firmly rooted in the market of hypoglycemic drugs. These include sitagliptin, vildagliptin, saxagliptin, alogliptin (registered in Japan), linagliptin (registered in the USA and Europe). At the same time, sitagliptin does not form covalent bonds with the active center of DPP-4 and acts less prolonged than other gliptins. Why was Sitagliptin chosen?

2)      Sitagliptin is prescribed to people in a daily dose of 100 mg. In the experiment, 100 mg / kg was administered once a day, i.e. approximately 50-100 times more than the daily dose for the treatment of patients. Where is the justification for the dose of Sitagliptin? (343) – “sitagliptin treatment (100 mg/kg/day) [22]” This link (doi:10.3390/jpm12040548) is not related to Sitagliptin!

3)     Considering that the excretion of DPP-4 inhibitors mainly occurs through the kidneys, then with the development of renal insufficiency, a dose adjustment of these drugs is required. DPP-4 inhibitors are allowed to be used at a therapeutic dose with a moderate decrease in GFR (50-80 mL/min/1.73m2), with GFR < 50 mL/min/1.73m2, the dose of drugs should be reduced by 50% or more [(1)European Medicines Agency (EMEA). Onglyza (saxagliptin)—European public assessment report (EPAR)—CHMP Assessment Report. Available from URL: http://www.emea.europa.eu/humandocs/PDFs/EPAR/onglyza/H-1039-en6.pdf. Accessed 5 July 2010.; (2) Bergman A.J., Cote J., Yi B. et al. Effect of renal insufficiency on the pharmacokinetics of sitagliptin, a dipeptidyl peptidase-4 inhibitor. Diabetes Care 2007; 30: 1862-1864.]. Why were Sitagliptin doses toxic to the kidneys used in such cases?

4)     Materials and Methods (n = 6 in each group); Table 1., Figure 2-3 (n = 8 samples/group)?

5)     (334-335) Diabetes was confirmed when the blood glucose 334 levels were > 20 mmol/L (200 mg/dl). A more accurate and standard method of verifying diabetes is the determination in the blood: HbA1c (Glycated Hemoglobin). This method is also used in experiments on rats (DOI: 10.1080/09712119.2018.1544902; DAY: https://doi.org/10.3126/ajms.v3i3.4816 ). Why didn't the authors use this method?

6)     (329-333) “The rats were subjected to 24-h fasting. Afterward, a single dose of STZ (30 mg/kg) 329 [21] diluted in 0.1 M citrate buffer (pH 4.5) immediately before injection was administered intraperitoneally to induce type 2 diabetes in the rats. Seventy-two hours after the STZ 331 injection, a drop of blood was collected from the tail vein, and random blood glucose lev-332 els were measured using the Accu-Chek Advantage II blood glucose monitoring system” The source [21] does not have open access, so it is difficult to understand the validity of the scheme used. Most other studies suggest different schemes for using STZ (usually 65 mg/kg):

Masiello P, Broca C, Gross R, Roye M, Manteghetti M, Hillaire-Buys D, Novelli M, Ribes G. Experimental NIDDM: development of a new model in adult rats administered streptozotocin and nicotinamide. Diabetes. 1998; 47(2):224-9. doi: 10.2337/diab.47.2.224. PMID: 9519717.

Zhu L, Han J, Yuan R, Xue L, Pang W. Berberine ameliorates diabetic nephropathy by inhibiting TLR4/NF-κB pathway. Biol Res. 2018 Mar 31;51(1):9. doi: 10.1186/s40659-018-0157-8. PMID: 29604956; PMCID: PMC5878418.

Albasher G, Alkahtani S, Al-Harbi LN. Urolithin A prevents streptozotocin-induced diabetic cardiomyopathy in rats by activating SIRT1. Saudi J Biol Sci. 2022 Feb;29(2):1210-1220. doi: 10.1016/j.sjbs.2021.09.045. Epub 2021 Sep 17. PMID: 35241966; PMCID: PMC8865018.

Gui L, Wang F, Hu X, Liu X, Yang H, Cai Z, Qi M, Dai C. Epigallocatechin Gallate Protects Diabetes Mellitus Rats Complicated with Cardiomyopathy through TGF-β1/JNK Signaling Pathway. Curr Pharm Des. 2022;28(33):2758-2770. doi: 10.2174/1381612828666220902115437. PMID: 36173051.

Ajayi AM, Badaki V, Adebayo OG, Ben-Azu B. Plukenetia conophora seed oil ameliorates streptozotocin-induced hyperglycaemia and oxidative stress in rats. Biomarkers. 2022 May;27(3):240-246. doi: 10.1080/1354750X.2021.2024601. Epub 2022 Jan 4. PMID: 34964401.

Hao W, Li N, Mi C, Wang Q, Yu Y. Salidroside attenuates cardiac dysfunction in a rat model of diabetes. Diabet Med. 2022 Mar;39(3):e14683. doi: 10.1111/dme.14683. Epub 2021 Sep 15. PMID: 34467560.

Hao W, Li N, Mi C, Wang Q, Yu Y. Salidroside attenuates cardiac dysfunction in a rat model of diabetes. Diabet Med. 2022 Mar;39(3):e14683. doi: 10.1111/dme.14683. Epub 2021 Sep 15. PMID: 34467560.

Lu CH, Ou HC, Day CH, Chen HI, Pai PY, Lee CY, Chen RJ, Chang RL, PadmaViswanadha V, Hsieh DJ, Huang CY. Deep sea minerals ameliorate diabetic-induced inflammation via inhibition of TNFα signaling pathways. Environ Toxicol. 2020 Apr;35(4):468-477. doi: 10.1002/tox.22882. Epub 2019 Dec 3. PMID: 31794124.

In this regard, provide a more detailed justification for your experiment.

Author Response

Reviewer#2

The topic is relevant, but I have a number of comments:

Response: Thank you very much.

1)     Representatives of the class of DPP-4 inhibitors (glyptins) are firmly rooted in the market of hypoglycemic drugs. These include sitagliptin, vildagliptin, saxagliptin, alogliptin (registered in Japan), linagliptin (registered in the USA and Europe). At the same time, sitagliptin does not form covalent bonds with the active center of DPP-4 and acts less prolonged than other gliptins. Why was Sitagliptin chosen?

Response: Thank you for your constructive comment. We chose sitagliptin because it is the most frequently used DPP-4 inhibitor in Saudi Arabia, based on a published systemic review (Al-Rubeaan et al. Physicians’ choices in the first-and second-line management of type 2 diabetes in the Kingdom of Saudi Arabia. Saudi Pharmaceut J. 2020; 28(3): p. 329-337. DOI: 10.1016/j.jsps.2020.01.014). The choice of sitagliptin over other DPP4 inhibitors in Saudi Arabia may depend on several factors, including efficacy, safety, cost, and availability. Sitagliptin has been shown to be an effective and well-tolerated DPP4 inhibitor for the treatment of type 2 diabetes and is associated with low-risk hypoglycemia and available in various dosage forms. Moreover, cost may be another factor when choosing sitagliptin over other glyliptins, as it is less expensive. The availability of sitagliptin in Saudi Arabia may also play a role in its selection over other DPP4 inhibitors. The drug may be available in local pharmacies and hospitals, making it a convenient option for patients and healthcare providers.

One of the advantages of sitagliptin is that it does not form covalent bonds with the active center of DPP4, making it a reversible inhibitor of the enzyme. This means that sitagliptin binds to the enzyme in a non-permanent way, allowing the enzyme to regain its activity once the drug has been metabolized or eliminated from the body. By contrast, other DPP4 inhibitors form covalent bonds with the active center of DPP4, resulting in a more prolonged inhibition of the enzyme. The advantages of reversible inhibition of the enzyme, as mediated by sitagliptin, are that it may allow for a more rapid onest and offset of the drug’s action and may reduce the risk of accumulation and toxicity with long-term use. The reversible inhibition of DPP4 by sitagliptin and its low potential for drug interactions are other important advantages that may make it a favorable choice.

2)      Sitagliptin is prescribed to people in a daily dose of 100 mg. In the experiment, 100 mg / kg was administered once a day, i.e. approximately 50-100 times more than the daily dose for the treatment of patients. Where is the justification for the dose of Sitagliptin? (343) – “sitagliptin treatment (100 mg/kg/day) [22]” This link (doi:10.3390/jpm12040548)h is not related to Sitagliptin!

Response: Thank you for this notification and comment. We adjusted the cited references in the manuscript regarding the use of sitagliptin at a dose of 100mg/kg in rats (their citation numbers after revising all references have become (62, 63, line 356) instead of the previous citation number (22). The dose of sitagliptin was used based on previous studies. Sitagliptin (100 mg/kg/day) has been used in many experimental studies conducted on rat models of type 2 diabetes.

Lim et al. Effect of a dipeptidyl peptidase-IV inhibitor. Des-fluoro-sitagliptin, on neointimal formation after balloon injury in rats. Plos One. 2012; 7(4): p. e35007. DOI: 10.1371/journal.pone.0035007

Nileshraj et al. Study on synergistic effects of bromocriptine and sitagliptin in streptozotocin-induced diabetic rats. Journal Clin Diagnostic Res. 2021; 15(4). DOI: 10.7860/JCDR/2021/47643.14799

Forest et al. Characterization of the exocrine pancreas in the male Zucker diabetic fatty rat model of type 2 diabetes mellitus following 3 months of treatment with sitagliptin. Endocrinology. 2014; 155(3): p. 783-792. DOI: 10.1210/en.2013-1781

Our approach is consistent with that of other studies that used 100 mg/kg in rats and did not notice any toxicities or mortality associated with sitagliptin. Another important point to note is that the results of these preclinical studies cannot be extrapolated directly to humans due to anatomical and physiological differences between rats and humans.

In addition, the only toxicity noticed with high doses of sitagliptin was in repeat-dose toxicity studies performed in rats (up to 184 days). The maximum non-lethal dose was 500 mg/kg/day for rats (48 times the human exposure, based on the AUC). In rats, renal toxicity was observed at systemic exposure values 58 times higher than the human exposure levels, while the no-effect level was 19 times higher than the human exposure level. Since sitagliptin is virtually completely absorbed following an oral dose in rodents, the initial body burden of the drug is likely to be more directly related to the dosage on a mg/kg body weight basis than on a plasma AUC basis. This would result in more than an exposure 2 orders of magnitude greater in these animal studies than in humans, based on dosage (500 versus 2 mg/kg/day). Given the very high renal elimination rate of the drug, in part due to its active transport into the kidney, the resulting renal exposures in rodents compared to humans are also correspondingly greater than would be predicted based on plasma AUC margins. The histological changes indicate a relatively nonspecific cytotoxicity associated with these very high exposures. The specificity of the renal toxicity for rodents exposed to very high doses is also supported by the lack of any renal toxicity in dogs and rhesus monkeys treated within exposure margins of 26- to 28-fold compared to patients (although no toxicology studies with rhesus monkeys were presented in the Original Marketing Application; thus, a safety margin for this species cannot be determined). In view of the much lower clinical exposures and body burden of the drug in humans at the clinical dose and the lack of any biochemical changes indicative of renal toxicity in patients, these high-dose findings in rodents are not considered clinically relevant. )https://www.ema.europa.eu/en/documents/scientific-discussion/xelevia-epar-scientific-discussion_en.pdf(

3)     Considering that the excretion of DPP-4 inhibitors mainly occurs through the kidneys, then with the development of renal insufficiency, a dose adjustment of these drugs is required. DPP-4 inhibitors are allowed to be used at a therapeutic dose with a moderate decrease in GFR (50-80 mL/min/1.73m2), with GFR < 50 mL/min/1.73m2, the dose of drugs should be reduced by 50% or more [(1)European Medicines Agency (EMEA). Onglyza (saxagliptin)—European public assessment report (EPAR)—CHMP Assessment Report. Available from URL: http://www.emea.europa.eu/humandocs/PDFs/EPAR/onglyza/H-1039-en6.pdf. Accessed 5 July 2010.; (2) Bergman A.J., Cote J., Yi B. et al. Effect of renal insufficiency on the pharmacokinetics of sitagliptin, a dipeptidyl peptidase-4 inhibitor. Diabetes Care 2007; 30: 1862-1864.]. Why were Sitagliptin doses toxic to the kidneys used in such cases?

Response: Thank you for your comments and suggested studies to refer to. However, other previous publications that used sitagliptin (100mg/kg) in rat animal models did not report that sitagliptin at 100mg/kg caused kidney toxicity (Lim et al., 2012 [4]; Mansour et al., 2021 [5]; and Hausenloy et al., 2013 [6]).

  1. Wang, Q., et al., STAT3 Suppresses cardiomyocytes apoptosis in CVB3-induced myocarditis via survivin. Front Pharmacol, 2020. 11: p. 613883.
  2. Teimouri, M., et al., The role of protein tyrosine phosphatase 1B (PTP1B) in the pathogenesis of type 2 diabetes mellitus and its complications. J Physiol Biochem, 2022. 78(2): p. 307-322.
  3. Chen, Q., et al., TRIM18-regulated STAT3 signaling pathway via PTP1B promotes renal epithelial-mesenchymal transition, inflammation, and fibrosis in diabetic kidney disease. Front Physiol, 2021. 12: p. 709506.
  4. Lim, S., et al., Effect of a dipeptidyl peptidase-IV inhibitor, des-fluoro-sitagliptin, on neointimal formation after balloon injury in rats. PLoS One, 2012. 7(4): p. e35007.
  5. Mansour, S.M., et al., Protective effect of sitagliptin and whole-body γ-irradiation in diabetes-induced cardiac injury. Can J Physiol Pharmacol, 2021. 99(6): p. 676-684.
  6. Hausenloy, D.J., et al., Dipeptidyl peptidase-4 inhibitors and GLP-1 reduce myocardial infarct size in a glucose-dependent manner. Cardiovasc Diabetol, 2013. 12: p. 154.

Sitagliptin is cleared in humans primarily by renal excretion of the parent drug, while in rats it is cleared by both renal and biliary excretion, indicating that the pharmacokinetics differ in humans and rats. Single-dose studies conducted with sitagliptin in rats have indicated that the highest non-lethal dose was 2000mg/kg for females and 3000 mg/kg for males (271 and 182 times the human exposure, respectively, based on AUC. https://www.ema.europa.eu/en/documents/scientific-discussion/xelevia-epar-scientific-discussion_en.pdf. The renal function of the rats in the present study was monitored to ensure that the sitagliptin dose did not induce any renal complications.

Sitagliptin was also tested in Zucker diabetic fatty (ZDF) rats as a model of type 2 diabetic nephropathy to investigate its effects on metabolic profiles and kidney injury. Sitagliptin administration improved kidney function, decreased renal oxidative stress, and abolished renal damage, as determined by histological examination. The reno-protective effects of sitagliptin in this study could be at least partly attributed to its ability to ameliorate hyperglycaemia and serum triglycerides. (doi:10.1155/2011/162092)

In a recent clinical study, Goldshtein and coworkers compared the change in urinary albumin-to-creatinine ratio (UACR) in type 2 diabetic patients with albuminuria who were receiving sitagliptin to those who were receiving sulphonyl urea as an add-on to metformin monotherapy. Both sitagliptin and sulphonyl urea decreased albuminuria as an add-on therapy to metformin; however, sitagliptin resulted in greater reductions in albuminuria independent of glycemic control when compared to sulphonyl urea.

Goldshtein et al. Urinary excretion with sitagliptin compared to sulphonyl urea as add on to metformin in type 2 diabetes patients with albuminuria: A real–world evidence study. J Diabetes Complications. 2016; 30(7): p. 1354-1359. DOI:10.1016/j.jdiacomp.2016.05.012.

4)     Materials and Methods (n = 6 in each group); Table 1., Figure 2-3 (n = 8 samples/group)?

Response: We appreciate your notification. We agree that repeating some sample measurements is not a common practice in research. The reason for repeating some sample measurements in this study was to provide more confirmation of the data obtained, and we aimed to obtain reliable data while saving time and resources.

5)     (334-335) Diabetes was confirmed when the blood glucose 334 levels were > 20 mmol/L (200 mg/dl). A more accurate and standard method of verifying diabetes is the determination in the blood: HbA1c (Glycated Hemoglobin). This method is also used in experiments on rats (DOI: 10.1080/09712119.2018.1544902; DAY: https://doi.org/10.3126/ajms.v3i3.4816 ). Why didn't the authors use this method?

Response: Thank you for your comment and suggestion. It is true that HbA1c is a more sensitive and useful test for monitoring blood glucose control in humans, but in our experimental animals, we detected blood glucose levels because immediate results are obtained and the levels can be checked more frequently than HbA1c levels, which represent the average blood glucose levels over the previous two to three months. The detection apparatus and kits are also more readily available and less expensive. In addition, animal models of diabetes often involve different pathophysiological mechanisms than those observed in humans with diabetes. Thus, the use of HbA1c as an indicator of blood glucose control may not be as relevant or informative in animal models.

Here, we have included some references for published studies that have used the same method for the detection of blood glucose levels as we did.

Al Za`abi et al. The effect of metformin in diabetic and non-diabetic rats with experimently-induced chronic kidney disease. Biomolecules. 2021; 11(6): p. 814. DOI: 10.3390/biom11060814.

Shawky et al. The biological impacts of sitagliptin on the pancreas of a rat model of type 2 diabetes mellitus: Drug interactions with metformin. Biology. 2020; 9(1): p. 6. DOI: 10.3390/biology9010006.

Nileshraj et al. Study on synergistic effect of bromocriptine and sitagliptin in streptozotocin-induced diabetic rats. J Clin Diagnostic Res. 2021; 15(4). DOI: 10.7860/JCDR/2021/47643.14799.

6)     (329-333) “The rats were subjected to 24-h fasting. Afterward, a single dose of STZ (30 mg/kg) 329 [21] diluted in 0.1 M citrate buffer (pH 4.5) immediately before injection was administered intraperitoneally to induce type 2 diabetes in the rats. Seventy-two hours after the STZ 331 injection, a drop of blood was collected from the tail vein, and random blood glucose lev-332 els were measured using the Accu-Chek Advantage II blood glucose monitoring system” The source [21] does not have open access, so it is difficult to understand the validity of the scheme used. Most other studies suggest different schemes for using STZ (usually 65 mg/kg):

Masiello P, Broca C, Gross R, Roye M, Manteghetti M, Hillaire-Buys D, Novelli M, Ribes G. Experimental NIDDM: development of a new model in adult rats administered streptozotocin and nicotinamide. Diabetes. 1998; 47(2): p. 224-229. DOI: 10.2337/diab.47.2.224. PMID: 9519717.

Zhu L, Han J, Yuan R, Xue L, Pang W. Berberine ameliorates diabetic nephropathy by inhibiting TLR4/NF-κB pathway. Biol Res. 2018;51(1): p. 9. DOI: 10.1186/s40659-018-0157-8. PMID: 29604956; PMCID: PMC5878418.

Albasher G, Alkahtani S, Al-Harbi LN. Urolithin A prevents streptozotocin-induced diabetic cardiomyopathy in rats by activating SIRT1. Saudi J Biol Sci. 2022; 29(2): p.1210-1220. DOI: 10.1016/j.sjbs.2021.09.045. Epub 2021 Sep 17. PMID: 35241966; PMCID: PMC8865018.

Gui L, Wang F, Hu X, Liu X, Yang H, Cai Z, Qi M, Dai C. Epigallocatechin gallate protects diabetes mellitus rats complicated with cardiomyopathy through TGF-β1/JNK signaling pathway. Curr Pharm Des. 2022; 28(33): p. 2758-2770. DOI: 10.2174/1381612828666220902115437. PMID: 36173051.

Ajayi AM, Badaki V, Adebayo OG, Ben-Azu B. Plukenetia conophora seed oil ameliorates streptozotocin-induced hyperglycaemia and oxidative stress in rats. Biomarkers. 2022; 27(3): p. 240-246. DOI: 10.1080/1354750X.2021.2024601. Epub 2022 Jan 4. PMID: 34964401.

Hao W, Li N, Mi C, Wang Q, Yu Y. Salidroside attenuates cardiac dysfunction in a rat model of diabetes. Diabet Med. 2022; 39(3): p. e14683. DOI: 10.1111/dme.14683. Epub 2021 Sep 15. PMID: 34467560.

Hao W, Li N, Mi C, Wang Q, Yu Y. Salidroside attenuates cardiac dysfunction in a rat model of diabetes. Diabet Med. 2022; 39(3): p. e14683. DOI: 10.1111/dme.14683. Epub 2021 Sep 15. PMID: 34467560.

Lu CH, Ou HC, Day CH, Chen HI, Pai PY, Lee CY, Chen RJ, Chang RL, PadmaViswanadha V, Hsieh DJ, Huang CY. Deep sea minerals ameliorate diabetic-induced inflammation via inhibition of TNFα signaling pathways. Environ Toxicol. 2020; 35(4): p. 468-477. DOI: 10.1002/tox.22882. Epub 2019 Dec 3. PMID: 31794124.

In this regard, provide a more detailed justification for your experiment.

Response: We appreciate your constructive comment and suggestion. With regard to reference (21), we adjusted it and added more references that support the animal model that we used in this study. These are now given citation numbers (12, 61) instead of the previous citation number (21), after revising all references in this manuscript (line 343). In this study, we targeted type 2 diabetes since the percentage of patients with type 2 diabetes has been considerably growing. In this study, we used a high-fat diet and a single low dose of streptozotocin (STZ; 30mg/kg) to induce type 2 diabetes. This model was established as a nongenetic rat model of type 2 diabetes mellitus to mimic the pathology of the human disease. While we used the high-fat diet to induce insulin resistance, the low dose of STZ causes a mild impairment of β cell function. Accordingly, we generated the scheme (Figure 1) to represent our study’s experimental design and approach. We used the fat-high diet/STZ rat model according to previous studies (listed below):

Wang HJ, Shen W, Neng J, Wu T, Li YJ. Low-dose streptozotocin (STZ) combined with high energy intake can effectively induce type 2 diabetes through altering the related gene expression. Asia Pac J Clin Nutr. 2007;16: p.412.

Qian C, Zhu C, Yu W, Jiang X, Zhang F. High-fat diet/low-dose streptozotocin-induced type 2 diabetes in rats impacts osteogenesis and Wnt signaling in bone marrow stromal cells. PLoS One. 2015;10(8): p. e0136390.

Zhang S, Xu H, Yu X, Wu YI, Sui D. Metformin ameliorates diabetic nephropathy in a rat model of low-dose streptozotocin-induced diabetes. Exper Ther Med. 2017;14(1): p. 383-390.

Shawky et al. The biological impacts of sitagliptin on the pancreas of a rat model of type 2 diabetes mellitus: Drug interactions with metformin. Biology. 2020; 9(6). DOI:10.3390/biology9010006.

Wang et al. Sitagliptin ameliorates diabetic nephropathy by blocking TGF-β1/Smad signalling pathway. Int J Molec Med. 2018; 41: p. 2784-2792

Note that HFD-fed rats treated with just a single high dose of STZ show clear features of type 1 diabetes, such as hyperglycemia, insulin deficiency, drastic weight loss, and resistance to insulin-sensitizing therapeutics. However, when the STZ dose is changed from a single high dose to a single low dose or multiple lower doses of STZ, researchers tend to agree that the HFD/STZ rat is a suitable model of type 2 diabetes. (https://doi.org/10.1002/cpz1.78). Our study uses a combination of HFD and a low dose of STZ treatment to generate a rat model that mimics the natural history and metabolic characteristics of the common type 2 diabetes in humans. It is inexpensive, easy to develop, and most suited for studying the pathophysiology of type 2 diabetes. It is also useful for evaluating therapeutic compounds for the treatment of type 2 diabetes and its complications, including diabetic nephropathy.

Round 2

Reviewer 2 Report

I did not understand your answer to remark number 4: why in materials and methods are the number of animals in subgroups of 6, and in tables of 8? Shouldn't these numbers be the same?

Author Response

Reviewer#2 (Round#2)`s Comment:

I did not understand your answer to remark number 4: why in materials and methods are the number of animals in subgroups of 6, and in tables of 8? Shouldn't these numbers be the same?

Response: We highly appreciate your notification and comment.

Rebuttal:

We have utilized 6 animals per group and we used some technical replicates making total numbers per experiment is eight from the six biological samples. This has been highlighted in Table 1 and figure 2 &3 legends emphasizing the sample number of biological sample per group.

If you wish to delete these technical replicates from the analysis we can proceed with that.

Thank you for your guidance
